# Tensorized Discrete Multi-View Spectral Clustering

**Qin Li [1], Geng Yang [1,*], Yu Yun [1,2], Yu Lei [1,2] and Jane You [3]**

1   School of Software Engineering, Shenzhen Institute of Information Technology, Shenzhen 518172, China;
    liqin@sziit.edu.cn (Q.L.); yuyun@stu.xidian.edu.cn (Y.Y.); 22011110198@stu.xidian.edu.cn (Y.L.)
2   School of Telecommunications Engineering, Xidian University, Xi'an 710071, China
3   Department of Computing, The Hong Kong Polytechnic University, Hong Kong 100872, China;
    csyjia@comp.polyu.edu.hk
*   Correspondence: yangg@sziit.edu.cn

**Abstract:** Discrete spectral clustering directly obtains the discrete labels of data, but existing clustering methods assume that the real-valued indicator matrices of different views are identical, which is unreasonable in practical applications. Moreover, they do not effectively exploit the spatial structure and complementary information embedded in views. To overcome this disadvantage, we propose a tensorized discrete multi-view spectral clustering model that integrates spectral embedding and spectral rotation into a unified framework. Specifically, we leverage the weighted tensor nuclear-norm regularizer on the third-order tensor, which consists of the real-valued indicator matrices of views, to exploit the complementary information embedded in the indicator matrices of different views. Furthermore, we present an adaptively weighted scheme that takes into account the relationship between views for clustering. Finally, discrete labels are obtained by spectral rotation. Experiments show the effectiveness of our proposed method.

**Keywords:** multi-view; spectral clustering; weighted tensor nuclear norm

## 1. Introduction

Multi-view clustering is attracting more and more attention in artificial intelligence and pattern recognition due to the fact that multi-view data, which are everywhere in reality, include some useful complementary information for clustering [1–6]. It aims to partition multi-view data into several clusters via rationally leveraging the complementary information such that the data points in the same cluster overall have high similarity to each other. One of the most representative multi-view clustering techniques is graph-based multi-view clustering, which has good performance on arbitrary shaped clusters.

Graph-based multi-view clustering aims to obtain the view-consensus adjacency matrix or embedding, i.e., indicator matrix, by using different diffusion strategies. As we all know, it is an NP-hard problem to optimize the graph-based clustering model with the discrete constraint imposed on the labels. To get rid of this problem, an intuitive method is to learn an approximate real-valued indicator matrix instead of discrete labels. Co-regularized spectral clustering (Co-reg) [7] is one of the most classical methods. It leverages the minimum mean squared error to minimize the divergence between the soft indicator matrices of different views. However, it implicitly considers that all views are equally important for clustering, which is unreasonable in practical applications. To this end, many related multi-view clustering methods have been developed, such as adaptive weighted multi-view spectral clustering methods [8,9] and adaptive graph learning clustering methods [10–13]. Although having promising clustering results, all of the aforementioned methods need post-processing operation, e.g., *K*-means, to obtain the clusters of data, i.e., discrete labels. This cannot make sure that the learned real-valued indicator matrix is optimal for *K*-means, which leads to suboptimal performance. Furthermore, the initialization of the cluster centroid points has a great influence on the clustering performance of *K*-means, which leads to unstable results.

To get rid of the aforementioned disadvantage, a frequently used technique is to learn the view-consensus graph with the connected component constraint [14,15]. Although these methods directly obtain the clusters of data via the number of connected components and achieve good clustering results in the experiments, it is difficult to learn or manually select the rational parameter to ensure that the learned graph has the exact number of connected components in reality. Moreover, these methods cannot effectively utilize the complementary information embedded in the adjacency matrices of different views. Another common used method is to achieve the discrete labels by joint spectral rotation [16]. However, spectral embedding and spectral rotation are two separate processes in the model, which results in sub-optimal performance. Moreover, it cannot effectively utilize the complementary information hidden in indicator matrices of different views. To achieve a rational solution, a unified framework was proposed [17–19] for discrete clustering. This framework simultaneously optimizes spectral embedding and spectral rotation to achieve the discrete labels, but it, despite working for single-view data, cannot be directly applied to multi-view data, which are ubiquitous in artificial intelligence and pattern recognition. Based on this framework, some works extended it to multi-view clustering [20–22]. But all of them minimize the divergence between the adjacency matrices of views by minimum mean square error, which is a one-dimensional and pixel-by-pixel measurement model. Thus, they cannot exploit the spatial structure and complementary information of views. Moreover, they assume that the real-valued indicator matrices of different views are identical, which does not make any sense in practical applications.

Drawing inspiration from the fact that the difference between soft indicator matrices of different views contains some useful complementary information for clustering [9], and inspired by the advantage of the weighted tensor nuclear norm [23,24] that can leverage the complementary information hidden in views, we utilize the weighted tensor nuclear-norm regularizer on the third-order tensor, which consists of real-valued indicator matrices of views, to minimize the divergence between the indicator matrices of views. Thus, the learned view-consensus real-valued indicator matrix encodes the complementary information of the indicator matrices of views. Meanwhile, considering the fact that different views play different roles in final clustering, we present an adaptive weighted strategy, which explicitly takes into account the relationship between views for clustering. The highlights of our paper can be listed as follows:

- Our model makes spectral clustering collaborate with spectral rotation in a unified framework for multi-view clustering. Thus, it directly obtains a discrete label matrix without post-processing.
- Our method effectively encodes both the complementary information and discriminative information of indicator matrices of the views by using a weighted tensor nuclear norm regularizer.
- Our weighted scheme directly considers the relationship between views for clustering. Thus, the learned indicator matrix effectively encodes discriminative information. Extensive experimental results indicate the effectiveness and efficiency of our proposed algorithm.

## 2. Related Works

Spectral clustering has attracted intensive attention in the literature due to its good performance on arbitrary shaped clusters and good spectral theory. Since multi-view data can provide some important complementary information for clustering, most multi-view spectral clustering methods have been proposed [7,22,25], where Co-reg [7] is one of the most classical clustering methods. It achieves the consensus indicator matrix, which is shared by all views, via leveraging the minimum mean squared error, and it has good clustering results. Wen et al. [26,27] applied it for incomplete multi-view clustering and proposed a new method. But they neglected the effect of different views for clustering. To take advantage of the effect of views for clustering, Nie et al. [8] developed an auto-weighted multiple graph learning spectral clustering. When it adaptively learns weights, it ignores

the interaction between views. To tackle this problem, Zong et al. [28] leveraged spectral perturbation theory to adaptively update weights for all views and proposed weighted multi-view spectral clustering (WMSC). However, all of them do not effectively utilize the complementary information. Inspired by the advantage of the tensor nuclear norm based on tensor singular value decomposition (t-SVD) [1,24], Xu et al. [9] used the weighted tensor nuclear norm regularizer to minimize the divergence between the indicator matrices of views and presented tensor low-rank constraint co-regularized spectral clustering, which adaptively updates weights for different views. Li et al. [29] leveraged the tensor nuclear norm regularizer on the tensor, which is composed of the normalized indicator matrices of views. Tan et al. [30] proposed to employ the topology of the data to capture the data manifold, while exploring the consistency information between different views in the sample level. However, the performance of the aforementioned spectral clustering methods depends on the quality of the pre-defined adjacency matrices of views.

To adaptively learn adjacency matrices which effectively present the relationship of the corresponding view data, many methods have been developed. For example, Zhan et al. [10] developed graph learning-based multi-view clustering (MVGL) which simultaneously learns the graph, which is shared by all views, and spectral embedding, but it ignores the spatial geometric structure of the adjacency matrix. To effectively exploit the spatial structure embedded in each adjacency matrix, Xia et al. [31] presented robust multi-view spectral clustering (RMSC), which learns the view-consensus graph with low-rank constraint. Tang et al. [32] leveraged rank constraint to learn the view consensus adjacency matrix and then achieved an indicator matrix by using classical spectral clustering. To effectively exploit the relationship between the adjacency matrices of views, Tang et al. [33] developed a parameter-free graph learning model by leveraging cross-view graph diffusion. The learned adjacency matrices effectively encode the cluster structure and discriminative information of views. Wu et al. [34] imposed a t-SVD-based tensor nuclear norm constraint on the third-order tensor, which is composed of the adjacency matrices of views, to obtain the adjacency matrix. It effectively encodes the complementary information and discriminative information of data.

Although achieving promising clustering results, all of the aforementioned methods need post-processing such as $K$-means to obtain the clusters of data. This leads to suboptimal clustering results. Moreover, the clustering results of their methods are unstable. To overcome this disadvantage, many methods have been developed to directly solve the discrete labels of data. For example, Nie et al. [14] sorted the manifold learning technique to learn the view-consensus adjacency matrix, which has $K$-connected components, and presented multi-view learning with adaptive neighbors (MLAN). However, MLAN assumes that each point has the same neighbors in different views. This does not make sense in reality due to the fact that each view has unique properties of the object that other views do not have. To improve the quality of the adjacency matrix, Nie et al. [15] learned the view-shared adjacency matrix from the pre-defined adjacency matrices of views. Zhan et al. [35] leveraged the indicator matrices of different views to learn the view-shared graph with $K$-connected components and presented the multi-view consensus graph clustering (MCGC) method. These methods can directly obtain the clusters of data according to the number of connected components, but it is difficult to ensure that the learned graph has the exact number of connected components in reality. To avoid this problem, according to the relationship between the non-negative matrix decomposition and spectral clustering relationship [36], Shi et al. [37] integrated the non-negative constraint into the graph learning model. In addition, Yang et al. [38] proposed a concise multi-view clustering model in order to avoid post-processing, where directly represents the cluster structure of the data through the learned common shared graph. The motivations of the aforementioned graph-learning methods are effective, but they do not effectively mine the complementary information embedded in different views.

Another commonly used technique leverages spectral rotation to obtain the clusters of data. For example, Yu and Shi [16] leveraged spectral rotation technique to achieve the discrete labels of the single view when the spectral embedding were obtained. Motivated by this work, Tian et al. [39] extended it to multi-view clustering. But in these two methods, spectral embedding and spectral rotation are two separate processes, resulting in sub-optimal clustering performance results. To achieve good clustering results, spectral embedding and spectral rotation are integrated into a unified framework for discrete clustering; many different algorithms have been developed to solve this framework [17–19]. Recently, some methods integrated the prediction function into this framework to solve the out-of-sample problem [40]. This framework directly obtains the discrete labels of data and achieves good performance, but it, despite working for single-view data, cannot be directly applied to multi-view data which are ubiquitous in artificial intelligence and pattern recognition. To get rid of this limitation, some works extended this framework to multi-view clustering [20–22]. However, they used minimum mean squared error, which is one-dimensional and a pixel-by-pixel measurement method, to learn the view-shared adjacency matrix. Thus, they cannot exploit both the spatial structure and complementary information of views. Moreover, they all assume that the real-valued indicator matrices of different views are identical, which dose not make sense in reality. To address these limitations, inspired by the ability of tensors to fully mine complementary information between views for better performance [1,23,41], we integrate multi-view spectral embedding and spectral rotation into a unified framework and employ the weighted tensor nuclear norm to uncover complementary information and spatial structural relationships within the embedding matrices across different views. Simultaneously, we adopt a judicious weighting scheme that thoroughly considers the relationship between clustering and views, offering an effective algorithm for solving the discrete label matrix. Our approach, by effectively integrating complementary information from different views, enables the model to better capture the diversity and complexity of the data. It holds the potential to provide robust support for intelligent annotation systems, particularly in domains such as image recognition, bioinformatics, and social network analysis. This can contribute to the advancement of artificial intelligence in practical scenarios.

## 3. Notations

According to [23,42], in this paper, the third-order tensor is denoted by bold calligraphy letter, i.e., $\mathcal{H} \in \mathbb{R}^{n_1 \times n_2 \times n_3}$, and $\mathcal{H}^{(i)} \in \mathbb{R}^{n_1 \times n_2}$ denotes the $i$-th frontal slice of tensor $\mathcal{H}$. Bold upper-case letters are used to denote matrices, e.g., $\mathbf{H}$, bold lower-case letters represent vectors, e.g., the $j$-th column $\mathbf{h}_j$ of $\mathbf{H}$, and lower-case letters are used for elements, e.g., the element $h_{ijk}$ of tensor $\mathcal{H}$. The discrete Fast Fourier Transform (FFT) of tensor $\mathcal{H}$ along the third dimension is $\overline{\mathcal{H}} = fft(\mathcal{H}, [], 3)$, and $\mathcal{H} = ifft(\overline{\mathcal{H}}, [], 3)$, where $ifft(\cdot)$ is the inverse Fast Fourier Transform. $\mathbf{I}$ denotes an identity matrix. The trace of matrix $\mathbf{H}$ is represented by $tr(\mathbf{H})$. $\mathbf{H}^{\mathrm{T}}$ is the transpose of $\mathbf{H}$.

## 4. Proposed Method

### 4.1. Problem Formulation and Objective Function

Given data matrix $\mathbf{X} \in \mathbb{R}^{d \times N}$, which contains $K$ clusters, where $d$ is the dimension of each data, $N$ denotes the number of multi-view data. Let $\mathbf{L}$ denote the Laplacian matrix. Spectral clustering aims to partition data matrix $\mathbf{X}$ into $K$ clusters by solving the following objective function:

$$\min_{\mathbf{F}^T \mathbf{F} = \mathbf{I}} tr\left(\mathbf{F}^{\mathrm{T}} \mathbf{L} \mathbf{F}\right) \tag{1}$$

where $\mathbf{F} \in \mathbb{R}^{N \times K}$ denotes the indicator matrix of data.

After obtaining indicator matrix **F**, a discrete solution can be obtained using *K*-means. This results in the sub-optimal discrete solution due to the instability of the *K*-means. To solve the problem, discrete spectral clustering is presented [16–19], and its objective function is

$$\min_{\mathbf{F}^{\mathrm{T}}\mathbf{F}=\mathbf{I}} tr\left(\mathbf{F}^{\mathrm{T}}\mathbf{L}\mathbf{F}\right) + \beta\|\mathbf{F}\mathbf{R} - \mathbf{Y}\|_{\mathbf{F}}^2$$
$$\text{s.t.} \quad \mathbf{R}^{\mathrm{T}}\mathbf{R} = \mathbf{I}, \mathbf{Y} \in \text{Ind} \tag{2}$$

where $\mathbf{R} \in \mathbb{R}^{K \times K}$ is a rotation matrix, and $\mathbf{Y} \in \mathbb{R}^{N \times K}$ denotes discrete label matrix. $0 \leq \beta$ is a balance parameter.

Although the model (2) obtains a good discrete solution for clustering, it only focuses on single-view clustering, and a similar investigation for multi-view spectral clustering has been found to be lacking so far. Moreover, for multi-view data $\mathbf{X}^{(v)} \in \mathbb{R}^{d_v \times N}$, $(v = 1, 2, \ldots, m)$, $m$ is the number of views, and $d_v$ is the dimension of the *v*-th view data matrix $\mathbf{X}^{(v)}$. Each view usually characterizes the property of the same object that cannot be exploited in other views, and different properties have different effects for clustering. So, the difference between the indicator matrices of different views can help provide useful complementary information for clustering. Moreover, good clustering should require that the similarity between the learned soft indicator matrices of different views are high, and all of them are also close to the ideal solution. Combining the aforementioned insight analysis, we have

$$\min_{\mathbf{F}^{(v)}, \mathbf{R}, \mathbf{Y}} \|\mathcal{F}\|_{\omega, \circledast} + \sum_{v=1}^{m} \left\{ tr\left(\mathbf{F}^{(v)\mathrm{T}}\mathbf{L}^{(v)}\mathbf{F}^{(v)}\right) \right.$$
$$\left. + \beta \left\|\mathbf{F}^{(v)}\mathbf{R}^{(v)} - \mathbf{Y}\right\|_{\mathbf{F}}^2 \right\} \tag{3}$$
$$\text{s.t.} \quad \mathbf{F}^{(v)\mathrm{T}}\mathbf{F}^{(v)} = \mathbf{I}, \mathbf{R}^{\mathrm{T}}\mathbf{R} = \mathbf{I}, \mathbf{Y} \in \text{Ind}$$

where $\mathbf{F}^{(v)}$ denotes the indicator matrix of the *v*-th view, and $\mathbf{L}^{(v)}$ is the Laplacian matrix of the *v*-th view. Lateral slices of $\mathcal{F}$ are composed of $\mathbf{F}^{(v)}$, i.e., $\mathcal{F}(:, v, :) = \mathbf{F}^{(v)}$. Thus, the size of $\mathcal{F}$ is $N \times m \times K$, and $\|\mathcal{F}\|_{\omega, \circledast}$ denotes the weighted tensor nuclear norm of $\mathcal{F}$ [23], which is defined as

$$\|\mathcal{F}\|_{\omega, \circledast} = \sum_{i=1}^{K} \sum_{j=1}^{\min(N,m)} \omega_j * \sigma_j(\overline{\mathcal{F}}^{(i)}) \tag{4}$$

where $\overline{\mathcal{F}}^{(i)}$ represents the *i*-th frontal slice of the tensor $\mathcal{F}$, $\sigma_j(\overline{\mathcal{F}}^{(i)})$ is the *j*-th singular value of matrix $\overline{\mathcal{F}}^{(i)}$, and $\omega_j$ denotes the *j* element of weighted vector $\omega$. By concatenating the clustering indicator matrices from multiple views into a third-order tensor, information from different views can be integrated. Utilizing a weighted tensor nuclear norm on the third-order tensor, measuring each frontal slice along the third dimension provides in-depth insights into the multi-view feature tensor. This approach better explores the complementary information in multi-view data.

In Equation (3), it can be seen that all indicator matrices $\mathbf{F}^{(v)}$ ($v = 1, 2, \cdots, m$) are treated equally. It means that all views have the same contribution for clustering. This is unreasonable in practical applications. Like the aforementioned analysis, different views characterize different contents of the same object, and different contents usually have distinct effects for clustering. Unfortunately, the model (3) neglects this. Although some weighted schemes have been presented to consider the effect of different views, in most existing weighted schemes, either the hyperparameters need to be manually selected, or the adaptive weights are independent of each other. This affects the stability of the algorithm. To overcome this problem, we describe a new weighted scheme to take into account the effect of different indictor matrices. Thus, we revise the model (3) as

$$\min_{\substack{\mathbf{F}^{(v)},\alpha^{(v)} \\ \mathbf{R},\mathbf{Y},\lambda^{(v)}}} \|\mathcal{F}\|_{\omega,\circledast} + \sum_{v=1}^{m} \left\{ \alpha^{(v)} tr\left(\mathbf{F}^{(v)\mathrm{T}}\mathbf{L}^{(v)}\mathbf{F}^{(v)}\right) \right.$$

$$\left. + \frac{\beta}{\lambda^{(v)}}\left\|\mathbf{F}^{(v)}\mathbf{R}^{(v)} - \mathbf{Y}\right\|_{\mathbf{F}}^{2} \right\} \tag{5}$$

$$\text{s.t.} \quad \mathbf{F}^{(v)\mathrm{T}}\mathbf{F}^{(v)} = \mathbf{I}, 0 \le \alpha^{(v)} \le 1, \textstyle\sum_{v=1}^{m}\alpha^{(v)} = 1$$

$$\mathbf{R}^{(v)\mathrm{T}}\mathbf{R}^{(v)} = \mathbf{I}, \mathbf{Y} \in \mathrm{Ind}, 0 \le \lambda^{(v)} \le 1, \textstyle\sum_{v=1}^{m}\lambda^{(v)} = 1$$

where $\lambda^{(v)}$ and $\alpha^{(v)}$ reflect the importance of the $v$-th view for clustering. Through a meaningful optimization process, our method assigns higher weights to beneficial views, further enhancing the algorithm's performance.From Equation (5), it can be observed that the second term represents the adaptive relaxed spectral embedding model, and the third term corresponds to the adaptive spectral rotation model. We integrate multi-view spectral embedding and spectral rotation into a unified framework, eliminating the need for post-processing, and directly obtain a discrete label matrix.

In Equation (5), we have that $\mathbf{F}^{(v)}\mathbf{R}^{(v)}$ is an orthonormal matrix, i.e., $\left(\mathbf{F}^{(v)}\mathbf{R}^{(v)}\right)^{\mathrm{T}}\left(\mathbf{F}^{(v)}\mathbf{R}^{(v)}\right) = \mathbf{I}$, while $\mathbf{Y}$ is a column orthogonal matrix, i.e., $\mathbf{Y}^{T}\mathbf{Y} = diag[n_1, n_2, \cdots, n_K]$, where $n_i$ ($i = 1, \cdots, K$) denotes the number of data in the $i$-th cluster. Thus, it is unreasonable to make an orthogonal matrix approximate an orthonormal matrix. To avoid this problem and obtain a good discrete solution, inspired by spectral clustering [16], we denote $\mathbf{H} = \mathbf{D}^{\frac{1}{2}}\mathbf{Y}(\mathbf{Y}^{\mathrm{T}}\mathbf{D}\mathbf{Y})^{-\frac{1}{2}}$ by $\mathbf{Y}$ in Equation (5), where $\mathbf{D} = \sum_{v=1}^{m}\frac{1}{v}\mathbf{D}^{(v)}$, and $\mathbf{D}^{(v)}$ is a degree matrix corresponding to $\mathbf{L}^{(v)}$. It is easy to prove that $\mathbf{H}$ is an orthonormal matrix, i.e., $\mathbf{H}^{\mathrm{T}}\mathbf{H} = \mathbf{I}$. Thus, our final objective function is

$$\min_{\substack{\mathbf{F}^{(v)},\alpha^{(v)} \\ \mathbf{R},\mathbf{H},\lambda^{(v)}}} \|\mathcal{F}\|_{\omega,\circledast} + \sum_{v=1}^{m} \left\{ \alpha^{(v)} tr\left(\mathbf{F}^{(v)\mathrm{T}}\mathbf{L}^{(v)}\mathbf{F}^{(v)}\right) + \frac{\beta}{\lambda^{(v)}}\left\|\mathbf{F}^{(v)}\mathbf{R}^{(v)} - \mathbf{H}\right\|_{\mathbf{F}}^{2} \right\}$$

$$\text{s.t.} \quad \mathbf{F}^{(v)\mathrm{T}}\mathbf{F}^{(v)} = \mathbf{I}, 0 \le \alpha^{(v)} \le 1, \textstyle\sum_{v=1}^{m}\alpha^{(v)} = 1 \tag{6}$$

$$\mathbf{R}^{(v)\mathrm{T}}\mathbf{R}^{(v)} = \mathbf{I}, 0 \le \lambda^{(v)} \le 1, \textstyle\sum_{v=1}^{m}\lambda^{(v)} = 1$$

**Remark 1.** *According to the construction of tensor $\mathcal{F}$, which is shown in Figure 1, we have that, for tensor $\mathcal{F}$, the $i$-th frontal slice $\Delta^{(i)}$ is a matrix whose columns are composed of vectors $\mathbf{F}^{(v)}_{:,i}$ ($v = 1, 2, \cdots, m$), where $\mathbf{F}^{(v)}_{:,i}$ denotes the $i$-th column of indicator matrix $\mathbf{F}^{(v)}$, which characterizes the relationship between $\mathbf{X}^{(v)}$ and the $i$-th cluster. The purpose of multi-view clustering is that $\mathbf{F}^{(1)}_{:,i}, \mathbf{F}^{(2)}_{:,i}, \mathbf{F}^{(m)}_{:,i}$ are as similar as possible; ideally, they are exactly equal. Moreover, in practical applications, there is a large difference between the cluster structures of different views. Thus, the first term in the model (6), i.e., the tensor multi-rank minimization constraint on $\mathcal{F}$, can make sure that $\Delta^{(i)}$ has a spatial low-rank structure, which helps exploit the complementary information embedded in the inter-views and obtain the view-consensus indicator matrix.*

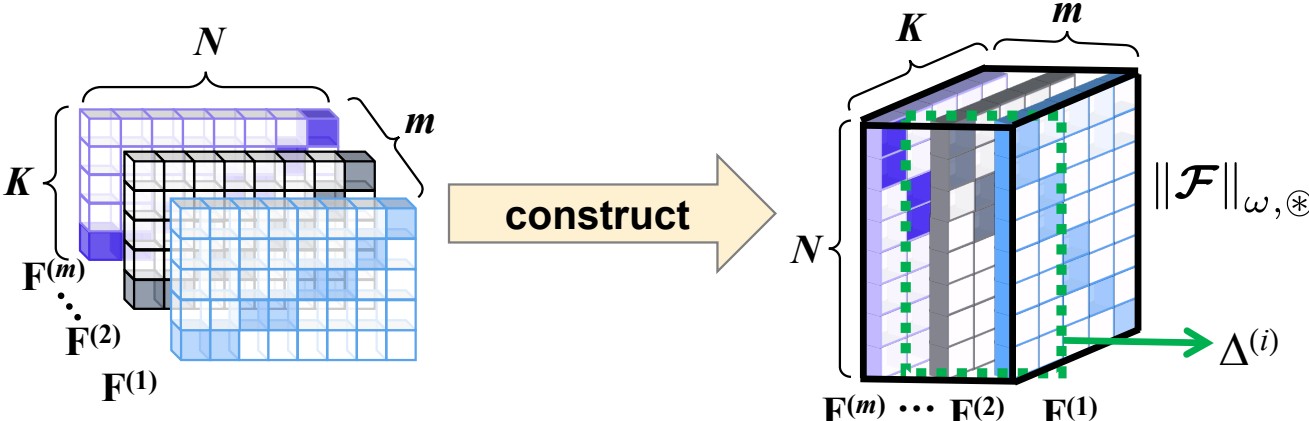

**Figure 1.** Construction of tensor $\mathcal{F} \in \mathbb{R}^{N \times m \times K}$. $\Delta^{(i)}$ denotes the *i*-th frontal slice of $\mathcal{F}$ ($i \in \{1,2,\dots,K\}$).

*4.2. Optimization*

Drawing inspiration from the augmented Lagrange multiplier (ALM) [43], we introduce an auxiliary tensor $\mathcal{J}$ instead of $\mathcal{F}$ in the model (6), and rewrite model (6) by solving the following minimization problem:

$$
\begin{aligned}
L(\mathcal{F}, \mathcal{J}, \alpha^{(v)}, \mathbf{R}^{(v)}, \mathbf{H}, \lambda^{(v)}) = \sum_{v=1}^{m} &\left\{ \alpha^{(v)} tr(\mathbf{F}^{(v)^{\mathrm{T}}} \mathbf{L}^{(v)} \mathbf{F}^{(v)}) \right. \\
&\left. + \frac{\beta}{\lambda^{(v)}} \|\mathbf{F}^{(v)} \mathbf{R}^{(v)} - \mathbf{H}\|_{\mathbf{F}}^2 \right\} \\
&+ \|\mathcal{J}\|_{\omega,\circledast} + \frac{\rho}{2} \|\mathcal{J} - (\mathcal{F} + \frac{\mathcal{G}}{\rho})\|_F^2
\end{aligned}
\tag{7}
$$

where tensor $\mathcal{G}$ denotes the Lagrange multiplier, and parameter $\rho > 0$ is an adaptive penalty factor.

To solve the model (7), it involves the following several subproblems, which can be alternately optimized.

- **Solving $\mathbf{R}^{(v)}$ with fixed $\mathbf{F}^{(v)}$ and $\mathbf{H}$.** In this case, each $\mathbf{R}^{(v)}$ can be solved independently. Thus, for the *v*-th rotation matrix $\mathbf{R}^{(v)}$, Equation (7) becomes

$$
\min_{\mathbf{R}^{(v)^{\mathrm{T}}} \mathbf{R}^{(v)} = \mathbf{I}} \|\mathbf{F}^{(v)} \mathbf{R}^{(v)} - \mathbf{H}\|_{\mathbf{F}}^2
\tag{8}
$$

By simple algebra, we have

$$
\begin{aligned}
\|\mathbf{F}^{(v)} \mathbf{R}^{(v)} - \mathbf{H}\|_F^2 &= tr((\mathbf{F}^{(v)} \mathbf{R}^{(v)})^{\mathrm{T}} (\mathbf{F}^{(v)} \mathbf{R}^{(v)})) \\
&\quad + tr(\mathbf{H}^{\mathrm{T}} \mathbf{H}) - 2tr((\mathbf{F}^{(v)} \mathbf{R}^{(v)})^{\mathrm{T}} \mathbf{H}) \\
&= \text{Constant} - 2tr((\mathbf{F}^{(v)} \mathbf{R}^{(v)})^{\mathrm{T}} \mathbf{H})
\end{aligned}
\tag{9}
$$

Then, the optimal solution of Equation (8) can be obtained by solving model (10):

$$
\max_{\mathbf{R}^{(v)^{\mathrm{T}}} \mathbf{R}^{(v)} = \mathbf{I}} tr(\mathbf{R}^{(v)^{\mathrm{T}}} \mathbf{F}^{(v)^{\mathrm{T}}} \mathbf{H})
\tag{10}
$$

Let the singular value decomposition (SVD) of $\mathbf{F}^{(v)^{\mathrm{T}}} \mathbf{H}$ be $\mathbf{F}^{(v)^{\mathrm{T}}} \mathbf{H} = \mathbf{U}^{(v)} \mathbf{S}^{(v)} \mathbf{V}^{(v)^{\mathrm{T}}}$, and according to Theorem 1, the optimal solution of the model (10) is

$$
\mathbf{R}^{(v)^*} = \mathbf{U}^{(v)} \mathbf{V}^{(v)^{\mathrm{T}}}
\tag{11}
$$

**Theorem 1** ([44]). *Let* $\mathbf{U}\Sigma\mathbf{V}^T$ *denote the compact singular value decomposition (SVD) of* $\mathbf{H}$*, then* $\mathbf{W} = \mathbf{U}\mathbf{V}^T$ *is the optimal solution of the following objective function:*

$$\max_{\mathbf{W}^T\mathbf{W}=\mathbf{I}} tr(\mathbf{W}^T\mathbf{H}) \tag{12}$$

- **Solving** $\mathcal{F}$ **with fixed** $\mathcal{J}, \alpha^{(v)}, \mathbf{R}^{(v)}, \mathbf{H}, \lambda^{(v)}$. Thus, Equation (7) becomes

$$\begin{aligned}
&\min_{\mathbf{F}^{(v)\mathrm{T}}\mathbf{F}^{(v)}=\mathbf{I}} \sum_{v=1}^{m} \alpha^{(v)} tr(\mathbf{F}^{(v)^\mathrm{T}}\mathbf{L}^{(v)}\mathbf{F}^{(v)}) \\
&\qquad + \frac{\beta}{\lambda^{(v)}} \|\mathbf{F}^{(v)}\mathbf{R}^{(v)} - \mathbf{H}\|_F^2 + \frac{\rho}{2}\|\mathcal{J} - (\mathcal{F} + \frac{\mathcal{G}}{\rho})\|_F^2 \\
&= \min_{\mathbf{F}^{(v)\mathrm{T}}\mathbf{F}^{(v)}=\mathbf{I}} \sum_{v=1}^{m} (\alpha^{(v)} tr(\mathbf{F}^{(v)^\mathrm{T}}\mathbf{L}^{(v)}\mathbf{F}^{(v)}) \\
&\qquad + \frac{\beta}{\lambda^{(v)}} \|\mathbf{F}^{(v)}\mathbf{R}^{(v)} - \mathbf{H}\|_F^2 + \frac{\rho}{2}\|\mathbf{A}^{(v)} - \mathbf{F}^{(v)}\|_F^2)
\end{aligned} \tag{13}$$

where $\mathbf{G}^{(v)}$ denotes the $v$-th frontal slice of tensor $\mathcal{F}$, $\mathbf{A}^{(v)} = (\mathbf{J}^{(v)} - \frac{\mathbf{G}^{(v)}}{\rho})$.

In Equation (13), we can rewrite the third term as

$$\begin{aligned}
\|\mathbf{A}^{(v)} - \mathbf{F}^{(v)}\|_F^2 &= tr((\mathbf{A}^{(v)})^\mathrm{T}\mathbf{A}^{(v)}) + tr((\mathbf{F}^{(v)})^\mathrm{T}\mathbf{F}^{(v)}) \\
&\quad - 2tr(\mathbf{F}^{(v)^\mathrm{T}}\mathbf{A}^{(v)}) \\
&= -2tr(\mathbf{F}^{(v)^\mathrm{T}}\mathbf{A}^{(v)}) + \text{Constant}
\end{aligned} \tag{14}$$

In Equation (13), each $\mathbf{F}^{(v)}$ can be solved independently. Moreover, substituting Equation (14) and Equation (9) into Equation (13), for the $v$-th $\mathbf{F}^{(v)}$, by simple algebra, we have

$$\begin{aligned}
&\min_{\mathbf{F}^{(v)\mathrm{T}}\mathbf{F}^{(v)}=\mathbf{I}} \alpha^{(v)} tr(\mathbf{F}^{(v)^\mathrm{T}}\mathbf{L}^{(v)}\mathbf{F}^{(v)}) \\
&\qquad - 2\frac{\beta}{\lambda^{(v)}} tr((\mathbf{F}^{(v)}\mathbf{R}^{(v)})^\mathrm{T}\mathbf{H}) - \rho tr(\mathbf{F}^{(v)^\mathrm{T}}(\mathbf{A}^{(v)})) \\
&= \min_{\mathbf{F}^{(v)\mathrm{T}}\mathbf{F}^{(v)}=\mathbf{I}} tr(\mathbf{F}^{(v)^\mathrm{T}}\alpha^{(v)}\mathbf{L}^{(v)}\mathbf{F}^{(v)}) \\
&\qquad - 2tr(\mathbf{F}^{(v)^\mathrm{T}}(\frac{\beta}{\lambda^{(v)}}\mathbf{H}\mathbf{R}^{(v)^T} + \frac{\rho}{2}\mathbf{A}^{(v)}))
\end{aligned} \tag{15}$$

**Theorem 2.** *Given model (16),*

$$\min_{\mathbf{F}^T\mathbf{F}=\mathbf{I}} tr(\mathbf{F}^T\mathbf{A}\mathbf{F}) - 2tr(\mathbf{F}^T\mathbf{P}) \tag{16}$$

*its optimal solution can be obtained by solving the model (17)*

$$\max_{\mathbf{F}^T\mathbf{F}=\mathbf{I}} tr(\mathbf{F}^T\tilde{\mathbf{A}}\mathbf{F}) + 2tr(\mathbf{F}^T\mathbf{P}) \tag{17}$$

*where* $\tilde{\mathbf{A}} = \lambda\mathbf{I} - \mathbf{A}$ *is a positive definite matrix.*

**Proof.** Multiply the model (16) by $-1$, and according to $\mathbf{F}^T\mathbf{F} = \mathbf{I}$, we easily obtain model (17). $\square$

According to Theorem 2, we can rewrite model (15) as model (18):

$$
\max_{\mathbf{F}^{(v)\mathrm{T}}\mathbf{F}^{(v)}=\mathbf{I}} tr(\mathbf{F}^{(v)\mathrm{T}}(\lambda\mathbf{I} - \alpha^{(v)}\mathbf{L}^{(v)})\mathbf{F}^{(v)})
$$
$$
+ 2tr(\mathbf{F}^{(v)\mathrm{T}}(\frac{\beta}{\lambda^{(v)}}\mathbf{H}\mathbf{R}^{(v)\mathrm{T}} + \frac{\rho}{2}\mathbf{A}^{(v)}))
$$

(18)

The problem of Equation (18) can be further written into

$$
\max_{\mathbf{F}^{(v)\mathrm{T}}\mathbf{F}^{(v)}=\mathbf{I}} \sum_{v=1}^{m} tr(\mathbf{F}^{(v)\mathrm{T}}(\mathbf{B}^{(v)}\mathbf{F}^{(v)} + 2\mathbf{C}^{(V)}))
$$

(19)

where $\mathbf{B}^{(v)} = (\lambda\mathbf{I} - \alpha^{(v)}\mathbf{L}^{(v)})$, $\lambda$ is an arbitrary constant to ensure that $\mathbf{B}^{(v)}$ is a positive definite matrix $\mathbf{C}^{(v)} = \frac{\beta}{\lambda^{(v)}}\mathbf{H}\mathbf{R}^{(v)\mathrm{T}} + \frac{\rho}{2}\mathbf{A}^{(v)}$.

In Equation (19), $\mathbf{B}^{(v)}\mathbf{F}^v$ ties up with the target variable $\mathbf{F}^v$, so Equation (19) cannot be solved directly. However, if we set $\mathbf{B}^{(v)}\mathbf{F}^v$ to be stationary, then Equation (19) can be easily solved by Theorem 1. Denote $\mathbf{U}\Sigma\mathbf{V}$ by the SVD of $\mathbf{B}^{(v)}\mathbf{F}^{(v)} + \mathbf{C}^{(v)}$, and according to Theorem 1, the solution of $\mathbf{F}^{(v)}$ is

$$
\mathbf{F}^{(v)*} = \mathbf{U}^{(v)}\mathbf{V}^{(v)\mathrm{T}}
$$

(20)

Algorithm 1 lists the pseudocode of solving $\mathbf{F}^{(v)}$.

---

**Algorithm 1** Solve $\mathbf{F}^{(v)}$

---

**Input**: The matrix $\mathbf{L}^{(v)}$, $\mathbf{F}^{(v)}$, $\mathbf{H}$, $\mathbf{R}^{(v)}$, and $\mathbf{A}^{(v)}$.
**Output**: $\mathbf{F}^{(v)}$, $\beta$.
1: **Initialize:** Compute $\mathbf{B}^{(v)} = \lambda\mathbf{I} - \alpha^{(v)}\mathbf{L}^{(v)}$, and $\mathbf{C}^{(v)} = \frac{\beta}{\lambda^{(v)}}\mathbf{H}\mathbf{R}^{(v)\mathrm{T}} + \frac{\rho}{2}\mathbf{A}^{(v)}$.
2: **while** not converge **do**
3: 　　Update $\mathbf{E}^{(v)} = \mathbf{B}^{(v)}\mathbf{F}^{(v)} + \mathbf{C}^{(v)}$;
4: 　　Calculate $\mathbf{U}\Sigma\mathbf{V}^{\mathrm{T}} = \mathbf{E}^{(v)}$ via compact SVD of $\mathbf{E}^{(v)}$;
5: 　　Update $\mathbf{F}^{(v)} = \mathbf{U}\mathbf{V}^{\mathrm{T}}$;
6: **end while**
7: **Return** the matrix $\mathbf{F}^{(v)}$.

---

- **Solving $\mathcal{J}$ with other fixed variables.** Now, $\mathcal{J}$ can be optimized by sub-problem (21):

$$
\min_{\mathcal{J}} \|\mathcal{J}\|_{\omega,\circledast} + \frac{\rho}{2}\|\mathcal{J} - (\mathcal{F} + \frac{\mathcal{G}}{\rho})\|_F^2
$$

(21)

To optimize model (21), we first introduce Theorem 3.

**Theorem 3** ([23]). *For $\mathcal{A} \in \mathbb{R}^{n_1 \times n_2 \times n_3}$, $l = \min(n_1, n_2)$, let $\mathcal{A} = \mathcal{U} * \mathcal{S} * \mathcal{V}^T$ (t-SVD). For*

$$
\underset{\mathcal{X}}{\mathrm{argmin}} \frac{1}{2}\|\mathcal{X} - \mathcal{A}\|_F^2 + \tau\|\mathcal{X}\|_{\omega,\circledast},
$$

(22)

*then, the optimal solution is*

$$
\mathcal{X}^* = \Gamma_{\tau*\omega}(\mathcal{A}) = \mathcal{U} * ifft(\mathbf{P}_{\tau*\omega}(\overline{\mathcal{A}})) * \mathcal{V}^T,
$$

(23)

*where $\overline{\mathcal{A}} = fft(\mathcal{A}, [], 3)$, $\mathbf{P}_{\tau*\omega}(\overline{\mathcal{A}})$ is a tensor, and $\mathbf{P}_{\tau*\omega}(\overline{A}^{(i)})$ is the i-th frontal slice of $\mathbf{P}_{\tau*\omega}(\overline{\mathcal{A}})$. $P_{\tau*\omega}(\mathbf{A}) = diag(\gamma_1, \gamma_2, \cdots, \gamma_l)$, $\gamma_i = \max(\mathbf{A}_{ii} - \tau * \omega_i, 0)$.*

According to Theorem 3, the solution of the model (21) is

$$
\mathcal{J}^* = \Gamma_{\frac{1}{\rho}*\omega}(\mathcal{F} + \frac{1}{\rho}\mathcal{G}).
$$

(24)

- **Solving $\mathcal{G}$ and $\rho$.** $\mathcal{G}$ can be updated by $\mathcal{G} = \mathcal{G} + \rho(\mathcal{F} - \mathcal{J})$. $\rho$ can be updated by $\rho = \rho\mu$, where $\mu$ is a positive number larger than 1.
- **Solving Y with fixed** $\lambda^{(v)}$, $\mathbf{F}^{(v)}$ **and** $\mathbf{R}^{(v)}$. In this case, the problem (7) becomes

$$\min_{\mathbf{Y} \in \text{Ind}} \sum_{v=1}^{m} \frac{\beta}{\lambda^{(v)}} \|\mathbf{F}^{(v)}\mathbf{R}^{(v)} - \mathbf{H}\|_{\mathbf{F}}^2 \tag{25}$$

Let $\mathbf{K}^{(v)} = \mathbf{F}^{(v)}\mathbf{R}^{(v)}$, then

$$\|\mathbf{F}^{(v)}\mathbf{R}^{(v)} - \mathbf{H}\|_{\mathbf{F}}^2 = tr(\mathbf{K}^{(v)\mathrm{T}}\mathbf{K}^{(v)}) + tr(\mathbf{H}^{\mathrm{T}}\mathbf{H}) \\ - 2tr(\mathbf{H}^{\mathrm{T}}\mathbf{K}^{(v)}) \tag{26}$$

It is easy to see that the first term in Equation (26) is not related to target variable $\mathbf{Y}$, and the second term is not also related to $\mathbf{Y}$ due to the fact $\mathbf{H}^{\mathrm{T}}\mathbf{H} = \mathbf{I}$. Thus, the optimal solution of the model (25) can be obtained by solving

$$\max_{\mathbf{Y} \in \text{Ind}} \sum_{v=1}^{m} tr(\frac{\beta}{\lambda^{(v)}}\mathbf{H}^{\mathrm{T}}\mathbf{K}^{(v)}) = \max_{\mathbf{Y} \in \text{Ind}} tr(\mathbf{H}\mathbf{P}^{\mathrm{T}}) \tag{27}$$

where $\mathbf{P} = \sum_{v=1}^{m} \frac{\beta}{\lambda^{(v)}}\mathbf{K}^{(v)}$.

Equation (27) suffers from the expensive time burden due to the calculation of $\mathbf{H} = \mathbf{D}^{\frac{1}{2}}\mathbf{Y}(\mathbf{Y}^{\mathrm{T}}\mathbf{D}\mathbf{Y})^{-\frac{1}{2}}$. To reduce the computational complexity, we propose a fast algorithm by Theorem 4.

**Theorem 4.** $\mathbf{H}$ *and* $\mathbf{Y}$ *have the same position of the non-zero element in each row. The non-zero element of the i-th row of* $\mathbf{H}$ *is* $\sqrt{\frac{d_i}{\mathbf{d}^T\mathbf{y}_j}}$.

**Proof.** For degree matrix $\mathbf{D}$, its degree vector is $\mathbf{d} = \mathbf{D}\mathbf{1}$, and $\mathbf{1}$ is a vector whose elements are all 1's. Thus, we have that the $i$-th row of $\mathbf{D}^{\frac{1}{2}}\mathbf{Y}$ is just the $i$-th row of $\mathbf{Y}$ multiplied by $\sqrt{d_i}$, i.e.,

$$(\mathbf{D}^{\frac{1}{2}}\mathbf{Y})_{ij} = \begin{cases} \sqrt{d_i}, & y_{ij} = 1 \\ 0, & \text{else} \end{cases} \tag{28}$$

where $d_i$ denotes the $i$-th element of vector $\mathbf{d}$.

Thus, $(\mathbf{Y}^{\mathrm{T}}\mathbf{D}\mathbf{Y})^{-\frac{1}{2}} = ((\mathbf{D}^{\frac{1}{2}}\mathbf{Y})^{\mathrm{T}}\mathbf{D}^{\frac{1}{2}}\mathbf{Y})^{-\frac{1}{2}}$ is a diagonal, and the $k$-th column of $\mathbf{Y}(\mathbf{Y}^{\mathrm{T}}\mathbf{D}\mathbf{Y})^{-\frac{1}{2}}$ is just the $k$-th column of $\mathbf{Y}$ multiplied by $(\mathbf{d}^{\mathrm{T}}\mathbf{y}_k)^{-\frac{1}{2}}$, where $\mathbf{y}_k$ is the $k$-th column of label matrix $\mathbf{Y}$.

According to the definition label matrix $\mathbf{Y}$, and combining the aforementioned analysis, we have that, for matrix $\mathbf{H}$, the $i$-th row $j$-th column element $h_{ij}$ is

$$h_{ij} = \begin{cases} \sqrt{\frac{d_i}{\mathbf{d}^{\mathrm{T}}\mathbf{y}_j}}, & y_{ij} = 1 \\ 0, & \text{else} \end{cases} \tag{29}$$

From Equation (29), we have that $\mathbf{H}$ and $\mathbf{Y}$ have the same position of non-zero element in each row. $\square$

Now, we consider how to optimize the model (27). Since each row of label matrix $\mathbf{Y}$ is independent, and each row has only one non-zero element, we can update each row of $\mathbf{Y}$ one by one. To update the $i$-th row of $\mathbf{Y}$, according to Theorem 4, we have

$$\mathbf{Y}_{ij} = \begin{cases} 1, & j = \arg\max_{k} tr(\mathbf{H}_{\mathbf{x}_i \in k}\mathbf{P}^{\mathrm{T}}) \\ 0, & \text{else} \end{cases} \tag{30}$$

where

$$\underset{\mathbf{x}_i \in k}{\mathbf{H}} = \mathbf{D}^{\frac{1}{2}} \underset{\mathbf{x}_i \in k}{\mathbf{Y}} \left( \underset{\mathbf{x}_i \in k}{\mathbf{Y}^{\mathrm{T}}} \mathbf{D} \underset{\mathbf{x}_i \in k}{\mathbf{Y}} \right)^{-\frac{1}{2}} \tag{31}$$

where $\underset{\mathbf{x}_i \in k}{\mathbf{Y}}$ means that by setting the *i*-th data as the *k*-th cluster, the others remain unchanged.

Note that according to Theorem 4, we have that the *i*-th row of matrix $\underset{\mathbf{x}_i \in k}{\mathbf{H}}$ is just the *i*-th row of matrix $\underset{\mathbf{x}_i \in k}{\mathbf{Y}}$ multiplied by a factor. Thus, $\underset{\mathbf{x}_i \in k}{\mathbf{H}}$ can be easily calculated by imitating Equation (29). Algorithm 2 lists the pseudo code for solving label matrix **Y**.

---

**Algorithm 2** Solving **Y**

---

**Input:d** $= \mathbf{D1} = [d_1, d_2, \cdots, d_n]$.
**Output**: **Y**.

1: **Initialize: H** $= 0$; label matrix **Y** whose each row has only one non-zero element 1;
2: **for** $i = 1, 2, \cdots, n$ **do**
3:      Calculate $\underset{\mathbf{x}_i \in k}{\mathbf{Y}}$;
4:      Calculate vector **g**. $g_i$ = j, if $y_{ij} = 1$;
5:      Calculate $\underset{\mathbf{x}_i \in k}{\mathbf{H}}$. Update non-zero element of the *i*-th row of **H** by $h_{i, g_i} = \sqrt{\dfrac{d_i}{\mathbf{d}^{\mathrm{T}} \mathbf{y}_{g_i}}}$, where
       $\mathbf{y}_{g_i}$ denotes the $g_i$-th column of $\underset{\mathbf{x}_i \in k}{\mathbf{Y}}$;
6:      Update *i*-row of **Y** according to Equation (30);
7: **end for**
8: **Return** the matrix **Y**.

---

- **Solving** $\lambda^{(v)}$. For the sake of a convincing description, let $\|\mathbf{F}^{(v)} \mathbf{R}^{(v)} - \mathbf{H}\|_{\mathbf{F}} = \zeta_v$ which is known. In this case, $\lambda^{(v)}$ can be solved by

$$\min_{\lambda^{(v)}} \sum_{v=1}^{m} \frac{1}{\lambda^{(v)}} \zeta_v^2, \quad \text{s.t.} \quad \sum_{v=1}^{m} \lambda^{(v)} = 1, \ \lambda^{(v)} \geq 0 \tag{32}$$

Due to $\sum_{v=1}^{m} \lambda^{(v)} = 1$, according to the Cauchy–Schwarz inequality, we have

$$\sum_{v=1}^{m} \frac{\zeta_v^2}{\lambda^{(v)}} = \left( \sum_{v=1}^{m} \frac{\zeta_v^2}{\lambda^{(v)}} \right) \left( \sum_{v=1}^{m} \lambda^{(v)} \right) \geq \left( \sum_{v=1}^{m} \zeta_v \right)^2 \tag{33}$$

In Equation (33), the equation holds if and only if $\sqrt{\lambda^{(v)}} \propto \frac{\zeta^{(v)}}{\sqrt{\lambda^{(v)}}}$. Moreover, the right-hand side in Equation (33) is a constant; thus, the optimal solution $\lambda^{(v)}(\forall v = 1, 2, \cdots, m)$ is

$$\lambda^{(v)} = \zeta_v / \sum_{v=1}^{m} \zeta_v \tag{34}$$

- **Solving** $\alpha^{(v)}$**.** According to [8], the optimal $\alpha^{(v)}$ is

$$\alpha^{(v)} = \frac{1}{2\sqrt{\left(\mathbf{F}^{(v)^{\mathrm{T}}} \mathbf{L}^{(v)} \mathbf{F}^{(v)}\right)}} \tag{35}$$

Finally, we summarize the aforementioned optimization procedure in Algorithm 3.

---

**Algorithm 3** Tensorized discrete multi-view spectral clustering

---

**Input**: Data matrices $\{\mathbf{X}^{(v)}\}_{v=1}^{m}$, hyperparameters: $K$, $\rho$, $\omega$ and $\beta$.
**Output**: The label $\mathbf{Y}$ of data.

1: **Initialize:** $\alpha^{(v)} = \frac{1}{m}$, $\lambda^{(v)} = \frac{1}{m}$, $\mathbf{S}^{(v)}$ according to [14];
2: Calculate degree matrix $\mathbf{D}^{(v)}$, $\mathbf{D} = \sum_{v=1}^{m} \frac{1}{v}\mathbf{D}^{(v)}$, $\mathbf{L}^{(v)} = \mathbf{I} - \mathbf{D}^{(v)-\frac{1}{2}}\mathbf{S}^{(v)}\mathbf{D}^{(v)-\frac{1}{2}}$ and $\mathbf{F}^{(v)}$
    according to standard spectral clustering, randomly initialize $\mathbf{Y}$;
3: **while** not converge **do**
4:    **for** $v = 1 : m$ **do**
5:       Calculate $\mathbf{R}^{(v)}$ by Equation (11);
6:       Calculate $\mathbf{F}^{(v)}$ by Algorithm 1;
7:       Calculate $\mathbf{J}^{(v)}$ by Equation (24);
8:    **end for**
9:    Calculate $\mathbf{Y}$ by Algorithm 2;
10:   Calculate $\mathcal{G}$ by $\mathcal{G} = \mathcal{G} + \rho(\mathcal{F} - \mathcal{J})$;
11:   **for** $v = 1 : m$ **do**
12:      Calculate $\lambda^{(v)}$ by Equation (34);
13:      Update $\alpha^{(v)}$ by Equation (35);
14:   **end for**
15:   Update $\rho$ by $\rho = \rho\mu$;
16: **end while**
17: **Return** the label matrix $\mathbf{Y}$ of data.

---

*4.3. Complexity Analysis*

The computational complexity of the proposed method mainly involves the four variables ($\mathbf{R}^{(v)}$, $\mathbf{F}^{(v)}$, $\mathbf{Y}$, and $\mathcal{J}$). Firstly, solving the $\mathbf{R}^{(v)}$-subproblem involves calculating the SVD decomposition of the $K \times K$ matrix, which is with the complexity of $\mathcal{O}(K^3)$. Secondly, the complexity of solving tensor $\mathbf{F}^{(v)}$ is $\mathcal{O}(t_1 K^2 N)$ because it involves the SVD decomposition of the $N \times K$ matrix, where $t_1$ denotes the number of iterations for solving $\mathbf{F}^{(v)}$. Thirdly, the computation of updating $\mathbf{Y}$ is $\mathcal{O}(K^2 N^2)$. Fourthly, solving the $\mathcal{J}$-subproblem involves calculating the 3D FFT and 3D inverse FFT of an $N \times m \times K$ tensor and $N$ SVDs of $K \times m$ matrices in the Fourier domain, both of which are with the complexity of $\mathcal{O}(2NmK\log(mK))$ and $\mathcal{O}(NKm^2)$. Since in multi-view scenarios we have $m \ll N$, and $K$ and $m$ are small constants, the main complexity of our proposed method approximately becomes $\mathcal{O}(K^2 N^2 + t_1 K^2 N + NK(2m\log(mK) + m^2))$ in each iteration. Despite a certain increase in complexity compared to some non-tensor algorithms, we endeavored to pursue a more comprehensive clustering performance through the incorporation of complementary information and spatial structure in multi-view data, coupled with the adoption of an adaptive weighting scheme.

**5. Converge Analysis**

**Theorem 5.** *Algorithm 1 has good convergence, i.e.,*

$$
\begin{aligned}
\sum_{v=1}^{m} &\Big( \frac{1}{\alpha_{t+1}^{(v)}} tr\big({\mathbf{F}_{t+1}^{(v)}}^{T} \mathbf{L}^{(v)} \mathbf{F}_{t+1}^{(v)}\big) + \frac{\rho}{2} \|\mathbf{A}^{(v)} - \mathbf{F}_{t+1}^{(v)}\|_F^2 \\
&+ \frac{\beta}{\lambda_{t+1}^{(v)}} \|\mathbf{F}_{t+1}^{(v)} \mathbf{R}^{(v)} - \mathbf{H}\|_{\mathbf{F}}^2 \Big) \\
\leq \sum_{v=1}^{m} &\Big( \frac{1}{\alpha_{t+1}^{(v)}} tr\big({\mathbf{F}_t^{(v)}}^{T} \mathbf{L}^{(v)} \mathbf{F}_t^{(v)}\big) + \frac{\rho}{2} \|\mathbf{A}^{(v)} - \mathbf{F}_t^{(v)}\|_F^2 \\
&+ \frac{\beta}{\lambda_{t+1}^{(v)}} \|\mathbf{F}_t^{(v)} \mathbf{R}^{(v)} - \mathbf{H}\|_F^2 \Big)
\end{aligned}
\tag{36}
$$

where $\mathbf{F}_{t+1}^{(v)}$ and $\mathbf{F}_t^{(v)}$ denote the optimal solution of Equation (15) in the $t+1$-th and $t$-th iterations in Algorithm 1, respectively.

**Proof.** For the $v$-th view, $\mathbf{B}^{(v)} = (\lambda\mathbf{I} - \alpha^{(v)}\mathbf{L}^{(v)})$ is positive definite, then we can rewrite $\mathbf{B}^{(v)} = \mathbf{U}^{\mathrm{T}}\mathbf{U}$ via Cholesky factorization. Moreover, we have

$$
\begin{aligned}
\|\mathbf{U}\mathbf{F}_{t+1}^{(v)} - \mathbf{U}\mathbf{F}_t^{(v)}\|_F^2 =\, & tr(\mathbf{F}_{t+1}^{(v)\,\mathrm{T}}\mathbf{B}^{(v)}\mathbf{F}_{t+1}^{(v)}) - \\
& 2tr(\mathbf{F}_{t+1}^{(v)\,\mathrm{T}}\mathbf{B}^{(v)}\mathbf{F}_t^{(v)}) + tr(\mathbf{F}_t^{(v)\,\mathrm{T}}\mathbf{B}^{(v)}\mathbf{F}_t^{(v)}) \\
\geq\, & 0
\end{aligned}
\tag{37}
$$

According to Algorithm 1, we have

$$
\begin{aligned}
& tr(\mathbf{F}_{t+1}^{(v)\,\mathrm{T}}\mathbf{B}^{(v)}\mathbf{F}_t^{(v)}) + tr(\mathbf{F}_{t+1}^{(v)\,\mathrm{T}}\mathbf{C}^{(v)}) \\
\geq\, & tr(\mathbf{F}_t^{(v)\,\mathrm{T}}\mathbf{B}^{(v)}\mathbf{F}_t^{(v)}) + tr(\mathbf{F}_t^{(v)\,\mathrm{T}}\mathbf{C}^{(v)})
\end{aligned}
\tag{38}
$$

where $\mathbf{C}^{(v)} = \frac{\beta}{\lambda^{(v)}}\mathbf{H}\mathbf{R}^{(v)\,\mathrm{T}} + \frac{\rho}{2}\mathbf{A}^{(v)}$.

Combining Equation (37) and Equation (38), we have

$$
\begin{aligned}
& tr(\mathbf{F}_{t+1}^{(v)\,\mathrm{T}}\mathbf{B}^{(v)}\mathbf{F}_{t+1}^{(v)}) + 2tr(\mathbf{F}_{t+1}^{(v)\,\mathrm{T}}\mathbf{C}^{(v)}) \\
\geq\, & tr(\mathbf{F}_t^{(v)\,\mathrm{T}}\mathbf{B}^{(v)}\mathbf{F}_t^{(v)}) + 2tr(\mathbf{F}_t^{(v)\,\mathrm{T}}\mathbf{C}^{(v)})
\end{aligned}
\tag{39}
$$

Substituting $\mathbf{B}^{(v)} = (\lambda\mathbf{I} - \alpha^{(v)}\mathbf{L}^{(v)})$ and $\mathbf{F}^{(v)\,\mathrm{T}}\mathbf{F}^{(v)} = \mathbf{I}$ into Equation (39), and by simple algebra, we have

$$
\begin{aligned}
& tr(\mathbf{F}_{t+1}^{(v)\,\mathrm{T}}\frac{1}{\alpha_{t+1}^{(v)}}\mathbf{L}^{(v)}\mathbf{F}_{t+1}^{(v)}) - 2tr(\mathbf{F}_{t+1}^{(v)\,\mathrm{T}}\mathbf{C}^{(v)}) \\
\leq\, & tr(\mathbf{F}_t^{(v)\,\mathrm{T}}\frac{1}{\alpha_{t+1}^{(v)}}\mathbf{L}^{(v)}\mathbf{F}_t^{(v)}) - 2tr(\mathbf{F}_t^{(v)\,\mathrm{T}}\mathbf{C}^{(v)})
\end{aligned}
\tag{40}
$$

Since all views are independent, we have

$$
\begin{aligned}
& \sum_{v=1}^m (tr(\mathbf{F}_{t+1}^{(v)\,\mathrm{T}}\frac{1}{\alpha_{t+1}^{(v)}}\mathbf{L}^{(v)}\mathbf{F}_{t+1}^{(v)}) - 2tr(\mathbf{F}_{t+1}^{(v)\,\mathrm{T}}\mathbf{C}^{(v)})) \\
\leq\, & \sum_{v=1}^m (tr(\mathbf{F}_t^{(v)\,\mathrm{T}}\frac{1}{\alpha_{t+1}^{(v)}}\mathbf{L}^{(v)}\mathbf{F}_t^{(v)}) - 2tr(\mathbf{F}_t^{(v)\,\mathrm{T}}\mathbf{C}^{(v)}))
\end{aligned}
\tag{41}
$$

Substituting $\mathbf{C}^{(v)} = \frac{\beta}{\lambda^{(v)}}\mathbf{H}\mathbf{R}^{(v)\,\mathrm{T}} + \frac{\rho}{2}\mathbf{A}^{(v)}$ into the model (40), by simple algebra, we have

$$
\begin{aligned}
& \sum_{v=1}^m tr(\mathbf{F}_{t+1}^{(v)\,\mathrm{T}}\frac{1}{\alpha_{t+1}^{(v)}}\mathbf{L}^{(v)}\mathbf{F}_{t+1}^{(v)}) \\
& - 2\sum_{v=1}^m tr(\mathbf{F}_{t+1}^{(v)\,\mathrm{T}}(\frac{\beta}{\lambda_{t+1}^{(v)}}\mathbf{H}\mathbf{R}^{(v)\,\mathrm{T}} + \frac{\rho}{2}\mathbf{A}^{(v)})) \\
\leq\, & \sum_{v=1}^m tr(\mathbf{F}_t^{(v)\,\mathrm{T}}\frac{1}{\alpha_{t+1}^{(v)}}\mathbf{L}^{(v)}\mathbf{F}_t^{(v)}) \\
& - 2\sum_{v=1}^m tr(\mathbf{F}_t^{(v)\,\mathrm{T}}(\frac{\beta}{\lambda_{t+1}^{(v)}}\mathbf{H}\mathbf{R}^{(v)\,\mathrm{T}} + \frac{\rho}{2}\mathbf{A}^{(v)}))
\end{aligned}
\tag{42}
$$

Combining Equation (42), Equation (14) and Equation (9), by simple algebra, we have that Equation (36) holds. □

Moreover, existing methods have demonstrated that [45] when the number of blocks is greater than or equal to 3, it is still an open problem to prove the convergence of ALM, which is leveraged in our proposed algorithm. Thus, we cannot prove the convergence of our proposed algorithm in theory. Fortunately, empirical evidence on a real dataset shows that our algorithm has a stable convergence property. We analyze the convergence on the Yale and MSRC-V1 datasets. Figure 2 lists the error $\|\mathbf{F}_{(k+1)}^{(v)} - \mathbf{J}_{(k+1)}^{(v)}\|_\infty$ versus the iteration number, where the *x-axis* and *y-axis* denote the iteration number and the corresponding error, respectively. It can be seen that the error decreases sharply and then becomes relatively stable in no more than 20 iterations. This indicates that our method has good convergence in real applications.

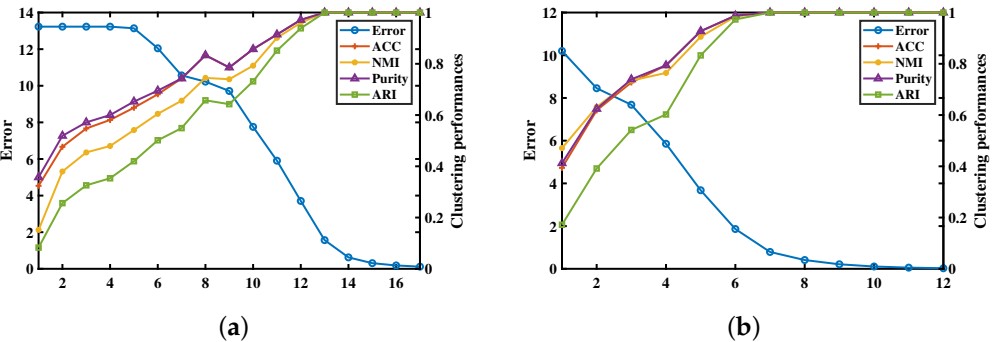

**Figure 2.** The convergence curves of our method on the MSRC-V1 and Yale datasets. (**a**) MSRC-V1. (**b**) Yale.

## 6. Experiment

We herein estimate the performance of our proposed method, which has been widely used in multi-view clustering, on five databases by using three standard clustering evaluation metrics, including Accuracy (ACC), Normalized Mutual Information (NMI), and the Purity and Adjusted Rand Index (ARI).

### 6.1. Experimental Setup

6.1.1. Datasets

In this subsection, we introduce five databases, which are used in the subsequent experiments:

- **Caltech101** (Cal-101) (https://tensorflow.google.cn/datasets/catalog/caltech101 (accessed on 10 September 2023)) [46] dataset has 101 classes. Like [9], we chose 441 samples with seven categories in our experiments. These samples have three views, i.e., 2560-dimensional (D) SIFT feature, 1160-D LBP features and 620-D HOG feature.
- The **MSRC-v1** (https://mldta.com/dataset/msrc-v1/ (accessed on 10 September 2023)) [47] dataset includes eight categories. Like [9], we chose 210 samples with seven categories in our experiments. These samples have five views, i.e., 24-D color moment, 512-D GIST feature, 576-D Histogram of Oriented Gradient, 254-D CENTRIST feature and 256-D LBP feature.
- The **Yale** (http://vision.ucsd.edu/content/yale-face-database (accessed on 10 September 2023)) dataset has 165 samples with 15 people. These samples have different conditions, such as occlusion changes, facial expression and with or without glasses. In the experiment, it has three different views, i.e., 3304-D LBP feature, 6750-D Gabor feature and 4096-D intensity feature.
- The **ORL** (http://www.uk.research.att.com/facedatabase.html (accessed on 10 September 2023)) dataset consists of 400 samples with 40 different people. These samples have different conditions, such as facial expression and occlusion changes. In the

experiment, we use three views, i.e., 6750-D Gabor feature, 4096-D intensity feature, and 3340-D LBP feature.

- **Scene-15** (https://www.kaggle.com/datasets/zaiyankhan/15scene-dataset (accessed on 10 September 2023)) [48] is a scene dataset with 15 different natural scenes. It contains 4485 samples with three different views, i.e., 1800-D PHOW featue, 1240-D CENTRIST feature and 1180-D PRI-CoLBP feature.
- The **ESP-GAME** (https://www.kaggle.com/datasets/parhamsalar/espgame (accessed on 10 September 2023)) dataset contains 11,032 samples over seven categories. It has two different views, each of them is 100 dimensions.

### 6.1.2. Comparisons

We herein compare our proposed method with 11 methods, including two representative tensor-based clustering methods (**T-SVD-MSC** [1] and **ETLMSC** [34]), three spectral clustering methods (**Co-Reg** [7], **LTCSPC** [9] and spectral clustering (**SC**) [49]), and six multi-graph fusion spectral clustering methods (**AMGL** [8], consistent and specific multi-view subspace clustering (**CSMSC**) [50], **MLAN** [14], **MCGC** [35], **RMSC** [31], and **MVGL** [10]). For our method, we tune the parameters $k$ (number of neighbors) from $[8, 9, 10]$, $\rho$ (penalty factor) from $[0.0001, 0.001, 0.01, 0.1]$, and $\beta$ from $[0.0001, 0.001, 0.01, 0.1]$, and the metric weight $\omega$ is set within the range of $(0, 100]$. Table 1 reports the values of the parameters on each dataset. To effectively estimate the performance our method with the aforementioned 11 methods, we repeat the experiments 50 times on each database in our paper.

**Table 1.** Parameters under best results on six datasets.

| Dataset | Yale | MSRC | Cal101 | ORL | Scene-15 | ESP-IMG |
|---------|------|------|--------|-----|----------|---------|
| $k$ | 8 | 8 | 8 | 8 | 8 | 80 |
| $\beta$ | 0.01 | 0.01 | 0.01 | 0.01 | 0.001 | 0.01 |
| $\omega$ | [1, 5, 0.1] | [1, 100, 10, 0.1, 5] | [5, 0.1, 1] | [1, 10, 100] | [1, 5, 100] | [10, 1] |

### 6.2. Experimental Results and Analysis

#### 6.2.1. Performance Comparison

Table 2 lists the ACC, NMI Purity and ARI of 11 methods on the aforementioned six datasets, where **SC-best** refers to the best results of **SC** among all views. From two tables, the following observations can be obtained:

- Multi-view clustering methods are overall superior to **SC** with the best performance. The reason may be that, compared with single-view representation, multi-view representation can provide some useful complementary information, and these multi-view methods make good use of these complementary information. In the Scene-15 database, except for **CSMSC**, non-tensor multi-view clustering methods are overall inferior to **SC**. The reason may be that there is a large difference between the views for clustering, and in these methods, weights for different views are independent and neglect the relationship between views for clustering. It also indicates that it is very important to design a suitable weighted scheme for improving multi-view clustering.
- Among the non-tensor clustering methods, the adaptive weighted multi-view spectral clustering method **AMGL** and coregularized spectral clustering **Co-reg** are overall inferior to the other methods. The reason may be that the performance results of **AMGL** and **Co-reg** heavily depend on the predefined graph, and it is difficult to manually select a suitable graph in real applications due to the complex distribution of data.
- Tensor-based clustering methods, such as **t-SVD-MSC**, **ETLMSC**, **LTCSPC** and our proposed method, are superior to the other methods. This is probably because tensor-based clustering methods effectively exploit the complementary information and spatial structure embedded in the graphs or affinity matrices of different views. It

also indicates that the difference between graphs or affinity matrices of different view helps provide useful complementary information for clustering.

**Table 2.** The clustering performance results on the Caltech101, MSRC-V1, Yale, ORL, SCENE-15 and ESP-GAME datasets.

| Dataset | Cal-101 | | | | MSRC-V1 | | | |
|---|---|---|---|---|---|---|---|---|
| Metric | ACC | NMI | Purity | ARI | ACC | NMI | Purity | ARI |
| SC-best | 0.545 ± 0.03 | 0.431 ± 0.02 | 0.624 ± 0.02 | 0.390 ± 0.02 | 0.663 ± 0.04 | 0.534 ± 0.02 | 0.674 ± 0.03 | 0.441 ± 0.03 |
| AMGL | 0.481 ± 0.03 | 0.345 ± 0.02 | 0.540 ± 0.02 | 0.184 ± 0.03 | 0.732 ± 0.04 | 0.669 ± 0.02 | 0.740 ± 0.02 | 0.543 ± 0.02 |
| CSMSC | 0.567 ± 0.00 | 0.480 ± 0.00 | 0.633 ± 0.00 | 0.395 ± 0.00 | 0.742 ± 0.01 | 0.597 ± 0.01 | 0.742 ± 0.01 | 0.532 ± 0.01 |
| MLAN | 0.587 ± 0.00 | 0.462 ± 0.00 | 0.655 ± 0.00 | 0.347 ± 0.00 | 0.743 ± 0.00 | 0.746 ± 0.00 | 0.805 ± 0.00 | 0.661 ± 0.00 |
| Co-Reg | 0.452 ± 0.01 | 0.283 ± 0.01 | 0.502 ± 0.01 | 0.234 ± 0.01 | 0.744 ± 0.02 | 0.632 ± 0.01 | 0.750 ± 0.01 | 0.635 ± 0.01 |
| RMSC | 0.529 ± 0.03 | 0.286 ± 0.02 | 0.565 ± 0.01 | 0.313 ± 0.02 | 0.742 ± 0.05 | 0.644 ± 0.03 | 0.764 ± 0.04 | 0.624 ± 0.02 |
| MCGC | 0.501 ± 0.00 | 0.376 ± 0.02 | 0.587 ± 0.00 | 0.301 ± 0.01 | 0.852 ± 0.00 | 0.724 ± 0.00 | 0.852 ± 0.00 | 0.749 ± 0.00 |
| MVGL | 0.483 ± 0.00 | 0.372 ± 0.00 | 0.571 ± 0.00 | 0.279 ± 0.00 | 0.852 ± 0.00 | 0.754 ± 0.00 | 0.852 ± 0.00 | 0.738 ± 0.02 |
| T-SVD-MSC | 0.828 ± 0.00 | 0.859 ± 0.00 | 0.868 ± 0.00 | 0.636 ± 0.00 | 0.999 ± 0.18 | 0.998 ± 0.40 | 0.999 ± 0.02 | 0.987 ± 0.02 |
| LTCSPC | 0.829 ± 0.01 | 0.822 ± 0.01 | 0.882 ± 0.01 | 0.643 ± 0.01 | 0.999 ± 0.00 | 0.999 ± 0.00 | 0.999 ± 0.00 | 0.989 ± 0.01 |
| ETLMSC | 0.642 ± 0.00 | 0.607 ± 0.00 | 0.739 ± 0.00 | 0.539 ± 0.01 | 0.995 ± 0.00 | 0.989 ± 0.00 | 0.995 ± 0.00 | 0.988 ± 0.01 |
| Ours | 0.832 ± 0.00 | 0.881 ± 0.00 | 0.912 ± 0.00 | 0.839 ± 0.00 | 1.00 ± 0.00 | 1.00 ± 0.00 | 1.00 ± 0.00 | 1.00 ± 0.00 |

| Dataset | Yale | | | | ORL | | | |
|---|---|---|---|---|---|---|---|---|
| Metric | ACC | NMI | Purity | ARI | ACC | NMI | Purity | ARI |
| SC-best | 0.556 ± 0.04 | 0.586 ± 0.04 | 0.567 ± 0.04 | 0.361 ± 0.04 | 0.727 ± 0.02 | 0.868 ± 0.01 | 0.762 ± 0.02 | 0.645 ± 0.03 |
| AMGL | 0.655 ± 0.02 | 0.654 ± 0.01 | 0.657 ± 0.02 | 0.394 ± 0.00 | 0.777 ± 0.02 | 0.883 ± 0.01 | 0.820 ± 0.02 | 0.633 ± 0.05 |
| CSMSC | 0.750 ± 0.00 | 0.776 ± 0.00 | 0.750 ± 0.00 | 0.615 ± 0.00 | 0.857 ± 0.02 | 0.935 ± 0.01 | 0.882 ± 0.01 | 0.813 ± 0.02 |
| MLAN | 0.641 ± 0.00 | 0.682 ± 0.00 | 0.641 ± 0.00 | 0.413 ± 0.00 | 0.684 ± 0.00 | 0.786 ± 0.01 | 0.735 ± 0.01 | 0.557 ± 0.01 |
| Co-Reg | 0.628 ± 0.01 | 0.660 ± 0.01 | 0.637 ± 0.02 | 0.521 ± 0.01 | 0.668 ± 0.01 | 0.824 ± 0.00 | 0.713 ± 0.01 | 0.600 ± 0.00 |
| RMSC | 0.703 ± 0.04 | 0.717 ± 0.02 | 0.710 ± 0.04 | 0.533 ± 0.03 | 0.747 ± 0.02 | 0.866 ± 0.01 | 0.760 ± 0.01 | 0.662 ± 0.02 |
| MCGC | 0.715 ± 0.00 | 0.677 ± 0.00 | 0.667 ± 0.00 | 0.534 ± 0.00 | 0.800 ± 0.00 | 0.895 ± 0.00 | 0.823 ± 0.01 | 0.679 ± 0.00 |
| MVGL | 0.709 ± 0.00 | 0.692 ± 0.00 | 0.709 ± 0.00 | 0.650 ± 0.00 | 0.765 ± 0.00 | 0.871 ± 0.00 | 0.815 ± 0.00 | 0.663 ± 0.00 |
| T-SVD-MSC | 0.932 ± 0.06 | 0.942 ± 0.05 | 0.932 ± 0.06 | 0.946 ± 0.03 | 0.962 ± 0.01 | 0.988 ± 0.00 | 0.973 ± 0.00 | 0.958 ± 0.01 |
| LTCSPC | 0.976 ± 0.01 | 0.982 ± 0.45 | 0.979 ± 0.70 | 0.965 ± 0.00 | 0.989 ± 0.01 | 0.994 ± 0.00 | 0.983 ± 0.01 | 0.978 ± 0.00 |
| ETLMSC | 0.659 ± 0.04 | 0.693 ± 0.04 | 0.659 ± 0.04 | 0.500 ± 0.05 | 0.958 ± 0.02 | 0.988 ± 0.01 | 0.970 ± 0.02 | 0.959 ± 0.02 |
| Ours | 1.00 ± 0.00 | 1.00 ± 0.00 | 1.00 ± 0.00 | 1.00 ± 0.00 | 1.00 ± 0.00 | 1.00 ± 0.00 | 1.00 ± 0.00 | 1.00 ± 0.00 |

| Dataset | SCENE-15 | | | | ESP-GAME | | | |
|---|---|---|---|---|---|---|---|---|
| Metric | ACC | NMI | Purity | ARI | ACC | NMI | Purity | ARI |
| SC-best | 0.483 ± 0.03 | 0.456 ± 0.01 | 0.534 ± 0.02 | 0.328 ± 0.06 | 0.512 ± 0.01 | 0.367 ± 0.01 | 0.539 ± 0.01 | 0.245 ± 0.00 |
| AMGL | 0.417 ± 0.03 | 0.473 ± 0.04 | 0.438 ± 0.03 | 0.285 ± 0.03 | 0.526 ± 0.00 | 0.354 ± 0.00 | 0.526 ± 0.00 | 0.264 ± 0.00 |
| CSMSC | 0.597 ± 0.00 | 0.573 ± 0.00 | 0.641 ± 0.00 | 0.439 ± 0.00 | 0.437 ± 0.00 | 0.284 ± 0.00 | 0.445 ± 0.00 | 0.221 ± 0.00 |
| MLAN | 0.340 ± 0.03 | 0.381 ± 0.04 | 0.351 ± 0.03 | 0.167 ± 0.03 | 0.476 ± 0.01 | 0.384 ± 0.00 | 0.496 ± 0.00 | 0.200 ± 0.01 |
| Co-Reg | 0.487 ± 0.00 | 0.466 ± 0.00 | 0.530 ± 0.00 | 0.324 ± 0.00 | 0.466 ± 0.01 | 0.375 ± 0.01 | 0.469 ± 0.01 | 0.181 ± 0.01 |
| RMSC | 0.451 ± 0.02 | 0.451 ± 0.01 | 0.490 ± 0.02 | 0.292 ± 0.02 | 0.446 ± 0.01 | 0.309 ± 0.01 | 0.468 ± 0.01 | 0.221 ± 0.01 |
| MCGC | 0.424 ± 0.00 | 0.406 ± 0.00 | 0.483 ± 0.00 | 0.378 ± 0.00 | 0.419 ± 0.00 | 0.240 ± 0.00 | 0.414 ± 0.00 | 0.125 ± 0.00 |
| MVGL | 0.449 ± 0.00 | 0.443 ± 0.00 | 0.464 ± 0.00 | 0.356 ± 0.00 | 0.473 ± 0.00 | 0.322 ± 0.00 | 0.478 ± 0.00 | 0.214 ± 0.00 |
| T-SVD-MSC | 0.816 ± 0.02 | 0.848 ± 0.01 | 0.867 ± 0.01 | 0.783 ± 0.01 | 0.569 ± 0.00 | 0.409 ± 0.00 | 0.579 ± 0.00 | 0.356 ± 0.00 |
| LTCSPC | 0.869 ± 0.01 | 0.863 ± 0.00 | 0.879 ± 0.01 | 0.813 ± 0.00 | 0.987 ± 0.00 | 0.963 ± 0.01 | 0.987 ± 0.00 | 0.971 ± 0.01 |
| ETLMSC | 0.873 ± 0.00 | 0.887 ± 0.00 | 0.905 ± 0.00 | 0.842 ± 0.00 | 0.730 ± 0.02 | 0.744 ± 0.01 | 0.682 ± 0.02 | 0.640 ± 0.02 |
| Ours | 0.909 ± 0.00 | 0.924 ± 0.00 | 0.912 ± 0.00 | 0.887 ± 0.00 | 0.993 ± 0.00 | 0.979 ± 0.00 | 0.993 ± 0.00 | 0.983 ± 0.00 |

- Our proposed method is superior to the other clustering methods. This is probably because our proposed method explicitly considers the effect of different views for clustering, and the assigned weights for different views are related. Moreover, our method directly obtains a discrete label matrix for data, while the other methods require extra post-processing, which results in a sub-optimal discrete solution. It is worth noting that after adjusting the parameters to achieve optimal results, we observed that the standard deviation of our algorithm across all datasets is 0. This indicates that our algorithm exhibits excellent robustness.

- **LTCSPC** utilizes weighted tensor nuclear norm regularization to minimize the discrepancy between the view indicator matrices, making full use of the complementary information hidden in the views. Therefore, its clustering performance is better than the adaptive weighted multi-view spectral clustering method (**AMGL**) and the

adaptive graph learning clustering method (**MVGL**). But **LTCSPC** is inferior to our proposed method. It indicates that spectral rotation helps further improve the clustering performance results and is a reasonable scheme to obtain a discrete solution for spectral clustering.

- Our model achieved relatively good clustering accuracy on various types of multi-view datasets, including Caltech101, MSRC-v1, Yale, ORL, Scene-15, and ESP-GAME. This consistent high performance demonstrates the strong generalization ability of our model, enabling it to adapt to the characteristics of different datasets and produce robust clustering results.

### 6.2.2. Parameter Analysis

We analyze the effect of weighted vector $\omega$ for our method. Figure 3 plots the performance results (ACC, NMI, and purity) of our proposed method with varying weighted vector $\omega$ on the Cal101 and Yale datasets, where *x-axis* denotes $\omega$, and *y-axis* is the clustering performance results. On the Cal101 database, our algorithm achieved optimal performance with $\omega = [5, 0.1, 1]$. On the Yale database, the optimal performance was observed with $\omega = [1, 5, 0.1]$, both significantly outperforming the equal-weight configuration $\omega = [1, 1, 1]$. This is attributed to the noticeable differences in the singular values of tensor $\mathcal{F}$, indicating that they should not be considered equally important. But it is very difficult to manually select a suitable weighted vector $\omega$, which exploits the salient difference in real applications. We will study it in our future work.

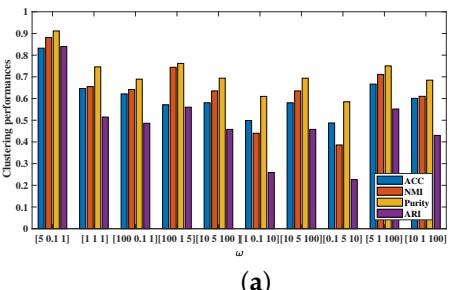
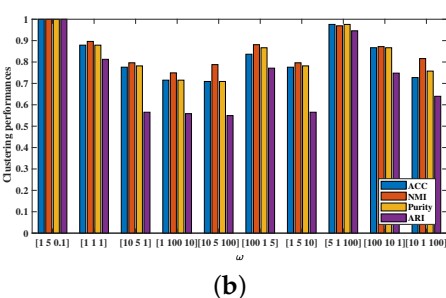

(**a**)                                                (**b**)

**Figure 3.** Analysis of the parameter $\omega$ on the MSRC-v1 and Yale datasets. (**a**) Cal101. (**b**) Yale.

We also analyze the effect of parameter $\beta$ for our method. Figure 4 shows the clustering performance versus $\beta$ on Yale and MSRC-V1 databases. When $\beta = 0$, our method becomes multi-view spectral clustering with a tensor low-rank constraint. From Figure 4, we have that our method has large fluctuation with varying $\beta$, and when $\beta = 0$, our method is inferior to the best performance with $\beta = 0.01$ in the Yale and MSRC-V1 databases. It also indicates that spectral rotation helps improve the clustering performance.

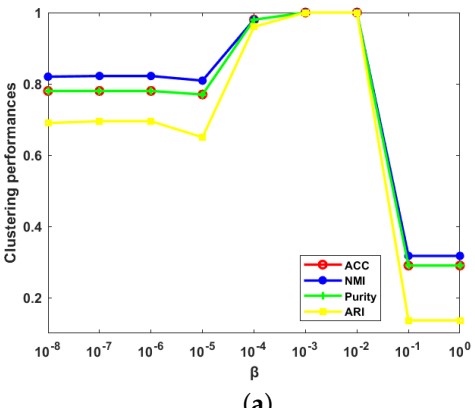
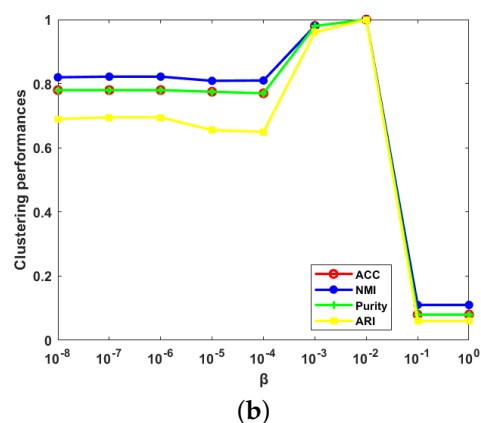

(**a**)                                                (**b**)

**Figure 4.** Analysis of the parameter $\beta$ on the MSRC-v1 and Yale datasets. (**a**) MSRC-v1. (**b**) Yale.

### 6.2.3. Ablation and Visualization

We analyze the ablation of parameters with $\alpha^{(v)} = 1$, $\lambda^{(v)} = 1$ and $\omega = [1, 1, 1]$ in Tables 3–5. It can be observed that setting the parameters to **1** represents a regular non-weighted scheme, treating each view matrix F(v) as equally important, and the results are worse. This emphasizes the importance of considering the distinctiveness of each view and assigning different weight values. This underscores the necessity of selecting an adaptive weighting scheme. What is more, Figure 5 shows the T-SNE of Scene-15, where, with the iteration growth, which means the results are near optimal, the data points exhibit a more compact distribution.

**Table 3.** Clustering results with/without adaptive weighting strategy $\alpha^{(v)}$, where ✗ means without an adaptive weighting strategy, and ✓ means with an adaptive weighting strategy. ↑ indicates that the higher the value of the indicator, the better the performance.

| Dataset | Yale | | | |
| --- | --- | --- | --- | --- |
| Method | ACC (↑) | NMI (↑) | Purity (↑) | ARI (↑) |
| ✗ | 0.994 | 0.993 | 0.994 | 0.987 |
| ✓ | 1.00 | 1.00 | 1.00 | 1.00 |
| Dataset | Cal-101 | | | |
| Method | ACC (↑) | NMI (↑) | Purity (↑) | ARI (↑) |
| ✗ | 0.658 | 0.741 | 0.764 | 0.601 |
| ✓ | 0.832 | 0.881 | f0.912 | 0.839 |

**Table 4.** Clustering results with/without adaptive weighting strategy $\lambda^{(v)}$, where ✗ means without an adaptive weighting strategy, and ✓ means with an adaptive weighting strategy. ↑ indicates that the higher the value of the indicator, the better the performance.

| Dataset | Yale | | | |
| --- | --- | --- | --- | --- |
| Method | ACC (↑) | NMI (↑) | Purity (↑) | ARI (↑) |
| ✗ | 0.994 | 0.993 | 0.994 | 0.987 |
| ✓ | 1.00 | 1.00 | 1.00 | 1.00 |
| Dataset | Cal-101 | | | |
| Method | ACC (↑) | NMI (↑) | Purity (↑) | ARI (↑) |
| ✗ | 0.773 | 0.791 | 0.848 | 0.698 |
| ✓ | 0.832 | 0.881 | 0.912 | 0.839 |

**Table 5.** Clustering results with/without weighted strategy $\omega$, where ✗ means without an adaptive weighting strategyv, and ✓ means with an adaptive weighting strategy. ↑ indicates that the higher the value of the indicator, the better the performance.

| Dataset | Yale | | | |
| --- | --- | --- | --- | --- |
| Method | ACC (↑) | NMI (↑) | Purity (↑) | ARI (↑) |
| ✗ | 0.715 | 0.749 | 0.715 | 0.559 |
| ✓ | 1.00 | 1.00 | 1.00 | 1.00 |
| Dataset | Cal-101 | | | |
| Method | ACC (↑) | NMI (↑) | Purity (↑) | ARI (↑) |
| ✗ | 0.646 | 0.655 | 0.746 | 0.512 |
| ✓ | 0.832 | 0.881 | 0.912 | 0.839 |

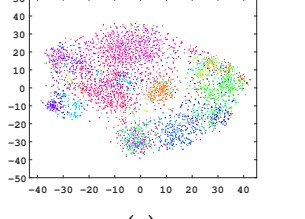 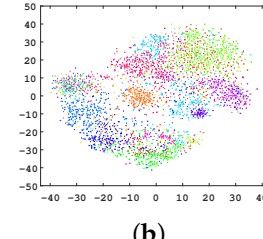 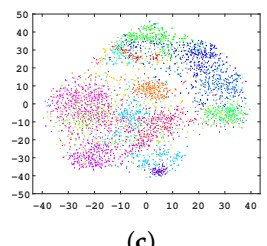 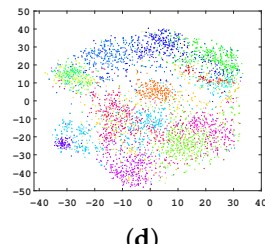

(**a**)         (**b**)         (**c**)         (**d**)

**Figure 5.** t-SNE visualizations on Scene dataset, where different colors represent different categories: (**a**) 3 epoch (Acc = 0.369), (**b**) 6 epoch (Acc = 0.592), (**c**) 9 epoch (Acc = 0.836), (**d**) 12 epoch (Acc = 0.909).

## 7. Conclusions

In this paper, we present tensorized discrete multi-view spectral clustering, which makes spectral embedding collaborate with spectral rotation in a unified framework. It leverages the weighted tensor nuclear norm regularizer to exploit the complementary information embedded in the indicator matrices of different views. Moreover, we present an adaptively weighted scheme that takes the relationship between views into consideration for clustering. Finally, an effective and efficient algorithm is proposed to solve the discrete label matrix. Extensive experimental results on different real-world datasets show that the proposed models outperform several multi-view methods. The weighted nuclear norm we employ relies on the significant differences in singular values among slices of the tensor formed by indicator matrices and the manual selection of weights. However, manually choosing an appropriate weighting vector is challenging in practice. We will study this in our future work.

**Author Contributions:** Conceptualization, Q.L.; methodology, Q.L. and G.Y.; writing—original draft preparation, Q.L.; writing—review and editing, Y.Y. and Y.L.; supervision, Q.L. and J.Y.; funding acquisition, Q.L. All authors have read and agreed to the published version of the manuscript.

**Funding:** This work was supported in part by Natural Science Foundation of Guangdong Province under Grant 2023A1515011845, and in part by 2022 Project of Shenzhen Education Science "14th Five Year Plan" under Grant zdzz22004.

**Data Availability Statement:** Publicly available datasets were analyzed in this study. These data can be found here: Caltech101, https://tensorflow.google.cn/datasets/catalog/caltech101 (accessed on 10 September 2023); MSRC-v1, https://mldta.com/dataset/msrc-v1/ (accessed on 10 September 2023); Yale, http://vision.ucsd.edu/content/yale-face-database (accessed on 10 September 2023); ORL, http://www.uk.research.att.com/facedatabase.html (accessed on 10 September 2023); Scene-15, https://www.kaggle.com/datasets/zaiyankhan/15scene-dataset (accessed on 10 September 2023); ESP-GAME, https://www.kaggle.com/datasets/parhamsalar/espgame (accessed on 10 September 2023).

**Conflicts of Interest:** The funders had no role in the design of the study; in the collection, analyses, or interpretation of data; in the writing of the manuscript, or in the decision to publish the results.

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
