# Peer review of "Tensorized Discrete Multi-View Spectral Clustering"

_electronics, doi:10.3390/electronics13030491_

Round 1

Reviewer 1 Report

Comments and Suggestions for Authors

I was unable to identify errors that could compromise the text or presentation of the proposal and results. The introduction justifies the topic and the authors' approach is innovative. Furthermore, the authors' algorithm was extensively tested and showed better results than conventional methods. Therefore, I recommend publishing the article as is.

Reviewer 2 Report

Comments and Suggestions for Authors

The paper needs a comprehensive revision before re-submission.

Comments on the Quality of English Language

This paper presents research on Tensorized Discrete Multi-view Spectral Clustering. I found this paper interesting because of its emphasis on a relevant topic. However, there are several notable shortcomings in this manuscript. Below, I mention the most crucial limitations of this article:

1) How does the proposed tensorized discrete multi-view spectral clustering model differ from existing methods, and what specific contributions does it make to address the identified limitations?

2) How is the integration of spectral embedding and spectral rotation achieved in the proposed unified framework, and what advantages does this integration offer for multi-view clustering?

3) Can authors elaborate on the rationale behind using the weighted tensor nuclear-norm regularizer on the third-order tensor and how it effectively exploits complementary information hidden in the views?

4) How does the proposed adaptive weighted scheme explicitly consider the relationship between views for clustering, and how does it improve the overall performance of the algorithm?

5) Are there any direct comparisons made with existing adaptive weighted multi-view spectral clustering methods and adaptive graph learning clustering methods? How does the proposed approach fare in terms of performance?

6) How robust are the experimental results in demonstrating the effectiveness of the proposed method, and are there specific datasets or scenarios where the method excels or struggles?

7) Please give the analysis regarding the computational efficiency of the proposed algorithm compared to existing methods, especially considering the potential complexity of tensorized operations.

8) How generalizable is the proposed model to different types of multi-view data, and are there any assumptions or limitations in the paper regarding the characteristics of the input data?

9) How does the proposed method address the sub-optimal performance resulting from the separation of spectral embedding and spectral rotation, especially concerning the exploitation of spatial structure and complementary information?

10) Please give a sensitivity analysis or discussion on the impact of different parameters in the proposed model, such as those related to the weighted tensor nuclear-norm regularizer or the adaptive weighted scheme.

11) Please add a detail in the research paper about how well the proposed method aligns with practical applications in artificial intelligence and pattern recognition, and are there insights into potential real-world use cases where the model might be particularly beneficial?

Reviewer 3 Report

Comments and Suggestions for Authors

The paper can be improved by addressing the following comments:

1- Why the coefficient lambda_v is in the denominator. Should it be in the numerator.

2- In Eq. (4), the notation F bar (i)  needs an explanation (which slice in FFT tensor)???

3- Equation (35) needs to be rectified:  1/    2 sqrt  ( Trace ( FT L F))

4- In Equations   (23) (24) Pw,t and Gamma_w  are not defined.

5- Check the first  mathematical expression at line 265. Is it  ifft or fft ??

6- The  references do not have many recent works. It is recommended to update this list with more recent papers.

7- Add the strengths and weaknesses of the scheme.

Round 2

Reviewer 2 Report

Comments and Suggestions for Authors

The authors have skillfully integrated the feedback provided earlier point-by-point, leading to significant improvements in the paper. The revised content now presents a more coherent and refined argument, thereby enhancing the overall quality of the research. I strongly recommend accepting the paper for publication, given the significant enhancements made during the revision process.